# Genetic Analysis of Multiple Primary Malignant Tumors in Women with Breast and Ovarian Cancer

**DOI:** 10.3390/ijms24076705

**Published:** 2023-04-04

**Authors:** Alina Savkova, Lyudmila Gulyaeva, Aleksey Gerasimov, Sergey Krasil’nikov

**Affiliations:** 1Federal Research Center of Fundamental and Translational Medicine, Novosibirsk 630117, Russia; 2V. Zelman Institute for the Medicine and Psychology, Novosibirsk State University, Novosibirsk 630090, Russia; 3E. Meshalkin National Medical Research Center of Ministry of Health of Russian Federation, Novosibirsk 630055, Russia; 4Novosibirsk Region Clinical Oncology Center, Novosibirsk 630108, Russia

**Keywords:** multiple primary malignant neoplasias, breast and ovary cancer syndrome, targeted genomic sequencing

## Abstract

Familial cancer syndromes, which are commonly caused by germline mutations in oncogenes and tumor suppressor genes, are generally considered to be the cause of primary multiple malignant neoplasias (PMMNs). Using targeted genomic sequencing, we screened for eight germline mutations: *BRCA1* 185delAG, *BRCA1* T300G, *BRCA1* 2080delA, *BRCA1* 4153delA, *BRCA1* 5382insC, *BRCA2* 6174delT, *CHEK2* 1100delC, and *BLM* C1642T, which provoke the majority of cases of hereditary breast and ovary cancer syndrome (HBOC), in genomic (blood) DNA from 60 women with PMMNs, including breast (BC) and/or ovarian cancer(s) (OC). Pathogenic allelic forms were discovered in nine samples: in seven instances, it was *BRCA1* 5382insC, and in the following two, *BRCA1* 4153delA and *BRCA1* T300G. The age of onset in these patients (46.8 years) was younger than in the general Russian population (61.0) for BC but was not for OC: 58.3 and 59.4, correspondingly. There were invasive breast carcinomas of no special type and invasive serous ovarian carcinomas in all cases. Two or more tumors of HBOC-spectrum were only in five out of nine families of mutation carriers. Nevertheless, every mutation carrier has relatives who have developed malignant tumors.

## 1. Introduction

PMMNs are defined as two or more histologically distinct malignancies that are not induced by metastasis, recurrence, or local spread within one individual.

If the second cancer was detected within 6 months after the first one, this means that PMMNs developed synchronously. Otherwise, the interval between them exceeds 6 months; it can be concluded that tumors occurred metachronously [1,2].

In the case of the development of three tumors, diagnoses are metachronous-metachronous, metachronous-synchronous, synchronous-metachronous, or synchronous-synchronous malignant neoplasms.

Nowadays, the absolute number of PMMNs and their proportion among newly diagnosed malignant neoplasias are on the rise due to improved diagnostics, the aging of the population, and the increased lifespan of cancer patients thanks to successful treatment methods [1,3]. Unfortunately, radiotherapy and certain types of pharmacotherapy also produce immunosuppressive and carcinogenic effects [1,4,5].

The pathogenesis of multiple and single tumors has similar mechanisms. Each person has a greater or lesser hereditary predisposition to the development of cancer, due to the presence in the genome of some germline oncogenic mutations with high, medium, and low degrees of penetrance. During life, under the influence of external (radiation or some carcinogenic substances) and internal (replication mistakes) factors, random somatic mutations occur in DNA to be added to germline ones. The combination of hereditary and acquired mutations in oncogenes and tumor suppressor genes initiates the process of carcinogenesis in the respective cells at an earlier or later age [2].

A decline in immune surveillance associated with aging, long-term stress, chronic inflammation, hormonal changes, certain infections, or immunosuppressive effects of treatment allows the tumor to continue its development. The whole sequence of events may repeat more than once in the same organism [4,5,6,7,8].

The development of metachronous tumors can be considered as an independent process. The simultaneous occurrence of malignant neoplasms may be associated with the action of the same triggering factor, which leads to the development of two or more tumors without any time gap [2,3].

In the Novosibirsk region, the incidence of PMNMs (34.6 per 100,000) exceeds the Russian average (24.8 per 100,000). This may stem from environmental conditions, in particular, high natural background radiation. Environmental influences, reproductive history, infections, behavioral and cultural factors (nutrition, smoking, alcohol consumption), working conditions, and potentially carcinogenic treatment—these risk factors are highly significant, as they lead to the occurrence of somatic mutations [1,4]. Nonetheless, a hereditary predisposition remains one of the main reasons for PMMNs’ development. 

Individuals carrying some germline mutations in oncogenes and tumor suppressor genes typically have a family history of cancer, develop a neoplasm at a young age, and often have PMMNs. Furthermore, each mutation leads to the occurrence of tumors of the corresponding spectrum; therefore, some hereditary oncological syndromes have been characterized (for example, HBOC syndrome, Li-Fraumeni syndrome, syndrome of multiple endocrine neoplasia, etc.).

There are hotspots in several genes, mutations in which lead to developing breast, ovarian, pancreatic, gastric, and prostate cancers, and hematological neoplasms [9,10].

If there is a history of BC and OC in the family, as a rule, its members are carriers of one of the following allelic variants: *BRCA1* 5382insC, *BRCA1* 4153delA, *BRCA1* 185delAG, *BRCA1* T300G, *BRCA1* 2080delA, *BRCA2* 6174delT, *CHEK2* 1100delC, *CHEK2* I157T, and *BLM* C1642T in Eastern Europe and in Russia [11,12,13,14,15].

All of these genes (and their corresponding proteins) are involved in the regulation of the cell cycle and may also influence the effects of sex hormones in hormone-dependent tissues [16,17,18,19,20].

In addition, a multiplicative interaction between several pathogenic mutations can also play a key role [21]. As regards hereditary *BRCA1/2* mutations, tumors develop in accordance with the classical two-hit model, whereas other mechanisms act in the case of *CHEK2* and *BLM* [22].

Thus, PMMNs are highly heterogeneous in their causes, triggering factors, and clinical manifestations (as well as malignancies in general). In order to identify the main patterns of pathology and prognosis, we analyzed all the necessary information about 60 female patients with PMMNs from the Novosibirsk region with BC, OC, or both: age of manifestation, interval between previous and subsequent malignancies, their localizations, treatments, and family cancer histories. Using targeted sequencing, we screened their genomic DNA for eight germline mutations most commonly associated with the development of the HBOC syndrome in this geographical area.

With the development and practical application of personalized therapies, such as poly(ADP-ribose)polymerase inhibitors (PARP inhibitors) for the treatment of *BRCA1/2*-mutation carriers, and other modern technologies, such as gene theranostics, it is possible to improve the survival and quality of life for patients predisposed to PMMNs [19,23].

## 2. Results

### 2.1. Patients’ Characteristics

In our group, there were 49 double primaries and 11 triple primaries.

Forty-eight women had BC, fifteen of them had non-metastatic ones in both mammary glands (in each individual instance, malignant tumors were histologically different or had a discrepancy in hormone-receptor status, or a time gap was at least 5 years), eight had BC and OC, and two had cancers in both mammary glands and ovarian one. Beyond that, 12 patients had OC with another malignant neoplasia(s). 

Contralateral BC, OC, and other reproductive tract malignancies (five endometrial carcinomas, five cervical cancers, four uterine sarcomas, one granulosa cell tumor, one borderline ovarian tumor, and one vaginal sarcoma), kidney and digestive tract malignancies (four stomach and two colorectal cancers) more commonly occurred in patients with BC (Figure 1a).

BC, other reproductive tract malignancies (four endometrial carcinomas and one cervical cancer), and thyroid cancers prevailed in patients with OC (Figure 1b).

The age of onset of the first tumor ranged from 27 to 76 years, with an average of 52.7 years. With regard to the second one, it was detected between 30 and 77 years, with a mean age of 58.7 years. The third tumor appeared between 39 and 84 years, with an average of 64.8 years (Figure 2). 

Synchronous tumors were developed in 12 patients, metachronous in 37 patients, metachronous-metachronous ones in 9 patients, and metachronous-synchronous in 2 patients. Thus, 23.3% of tumors progressed simultaneously (Figure 3).

Potentially carcinogenic treatments for the previous tumor were performed on 41 (85.4%) of 48 patients with metachronous tumors. In 20 cases, it was chemotherapy using substances that are carcinogenic to humans (Group 1 according to the classification of the International Agency for Research on Cancer (IARC)); in 1 case, treatment included probably carcinogenic to humans agents (Group 2A); in 8 cases, radiotherapy was applied; and in 13, complex treatment: carcinogenic substances (Group 1) along with radiotherapy.

Some of the therapeutic methods used lead to DNA damage and immunosuppression (alkylating agents and radiation); others interfere with the action of sex hormones (tamoxifen) [5,24].

We classified 28 family histories as not aggravated and 15 as aggravated by HBOC-spectrum malignancies. In 11 cases, there were at least two malignant tumors in relatives, but less than two of them belonged to the HBOC-spectrum (Figure 4).

### 2.2. Targeted Sequencing 

Pathogenic allelic variants were discovered in nine samples (Table 1). In seven cases, there was *BRCA1* 5382insC (rs80357906), in one case, *BRCA1* 4153delA (rs80357711), and in another one, *BRCA1* T300G (rs28897672). 

We consider it appropriate to give examples of mutation carriers’ family histories, paying attention to the fact that not all of them are aggravated by HBOC-spectrum malignancies (see Figure 5 and Figure 6 below).

## 3. Discussion

A combination of many factors leads to the development of PMMNs. However, some patients’ characteristics such as the age of tumor onset, family history of cancer, time gap between previous and subsequent malignancies, localizations, and potentially carcinogenic treatments, are most important for understanding the general patterns of pathology and prognosis.

Of the causes, the most significant is the carriage of oncogenic mutations with high and medium degrees of penetrance. Due to the contribution of such cases to the statistics, tumors in patients with PMMNs develop at a younger age than the average age of the cancer manifestation in the same geographical area.

So, the mean age of first (52.7 years) and even second (58.7 years) neoplasms in the group was lower than the age of cancer manifestation in Russia (63.9 years for women in 2019). The same trend is noted by other authors [25,26]. The mean age of BC for the first and subsequent tumors (64 malignancies, 56.1 years) is lower than the average in Russia (61.0 years). The mean age of OC (22 malignancies, 54.6 years) is also lower than the average in Russia (59.4 years). The difference is about five years.

The mean age of BC manifestation in *BRCA* mutation carriers was 46.8 years (nine cases). This result is consistent with data from a large study that included 3797 *BRCA1* mutation carriers. The average age was 40 years in that study [27]. Thus, BC in *BRCA1* mutation carriers develops 15–20 years earlier than in the general population. The mean age of OC onset (58.3 years, five cases) appears not to differ from that of the general population (59.4 in Russia in 2019). Other studies do not contradict our data [28].

In all examined patients, 23.3% of tumors had simultaneous progression. The rate of synchronous primary malignancies during the past ten years in Russia is 26.1–30.1. In the Novosibirsk region, its share is about 25%. In other countries, the proportion is nearly the same [3,4]. It seems probable that a single triggering factor can lead to the development of several tumors at the same time in a patient with a cancer predisposition. Sometimes it is a big problem to find an anticancer therapy that covers both cancer types without increased toxicity and pharmacological interactions [1]. In addition, many widely used methods have potentially carcinogenic effects.

An association between certain types of treatment (chemotherapy and radiotherapy) and the development of a subsequent metachronous tumor is proven. Such therapy increases the lifetime risk of leukemia, kidney cancer, and some other malignancies [4,29]. Radiotherapy increases the risk of thyroid cancer and subsequent malignancies of breast, bone, connective tissue, and lung at the area of exposure [1]. Two patients from the group developed leukemia several years after the complex treatment, and two developed kidney tumors. It is interesting to note that all of them have an aggravated family history, and potentially carcinogenic treatment is an additional risk factor in these cases.

The most common PMMNs are gender-specific [3]. The same proportion is observed for single malignancies. Sex steroid hormones act as growth factors. The peculiarities of the metabolism of these hormones and their action on the receptors in the tissues are associated with the presence of hereditary and acquired mutations in several genes (*BRCA1/2*, *CHEK2*, *CYP1A1*, *CYP19*, *SULT1*, and others), reproductive history, and hormone therapy use. 

Sex hormones and the expression of their respective receptors affect the growth of tumors of the ovaries, breast, reproductive tract, and colon [3,30]. Many studies indicate a relationship between the patient’s reproductive history and the development of tumors in these organs [1]. Of course, the role of sex hormones in carcinogenesis in the digestive tract is still debatable. However, the risk of colorectal cancer is higher in those patients who had tumors of the breast or reproductive tract [31], and the risk of BC is higher in patients with a history of gastric, colon, endometrial, or ovarian cancer [32]. In addition, the hormonal treatment of a primary BC increases the risk of these types of cancer [1].

The cumulative proportion of combinations with such tumors for the BC is 83.9%, and for the OC, 70.4%.

C. Frank et al. described that shared environmental risk factors (spousal risk) have less of an impact on familial risk than shared genes. The strongest correlation was observed in relatives for prostate, breast, and colorectal cancer. The authors note that many of the underlying genes are still unknown. Well-known mutations in highly penetrant genes explain a small proportion of the genetic basis of cancers, whereas familial aggregation has been suspected for almost all cancers [33,34].

An aggravated family history is considered as an additional basis for mutation screening in the cases of prostate, colon, endometrium, and ovary cancer [34]. However, the current guidelines for HBOC syndrome (the Mainstreaming Cancer Genetics (MCG) criteria) do not mention a family history of cancer as an important basis for testing, but only as an additional one [35].

Only five (1, 3, 6, 7, 8) out of nine family histories were aggravated by tumors of the HBOC spectrum. 

Nevertheless, every mutation carrier has relatives who have developed malignant tumor(s). Therefore, a non-aggravated family history does not guarantee the absence of pathogenic variants, and cancer-based MCG criteria appear to be more effective than previous ones [36].

All *BRCA1* mutation carriers, except patient 8, identify themselves with Russian nationality. Patient 8 is Ashkenazi Jewish but has a mutation *BRCA1* T300G classified as not typical in representatives of this nationality [36].

The carriage of each specific mutation leads to the development of the corresponding histological types of tumors. In all well-described cases, *BRCA* mutation carriers have had invasive breast carcinomas of no special type, also called invasive ductal carcinomas (grade II or III), and invasive serous ovarian carcinomas. These histological types of BC and OC develop in 80% and 67% of cases, respectively, in *BRCA1* mutation carriers [27]. Patients 4 and 7 also had non-HBOC tumors. Patient 4 developed serous endometrial carcinoma 12 years after treatment of BC with tamoxifen. However, not only tamoxifen but also germline *BRCA1* 5382insC may increase the risk of this histopathological subtype of uterine cancer [37]. Patient 7 developed uterine leiomyosarcoma at the age of 62, i.e., 30 and 12 years after the first and the second primary BCs, respectively. This patient had never been treated with tamoxifen, which could also exacerbate the risk of uterine sarcoma [24]. Therefore, the development of sarcoma in the patient may be associated with *BRCA1* 5382insC. This relationship was described by Laitman Y. et al. [37].

All of the newly identified carriers of mutations and their relatives received personal recommendations for surveillance. The possibilities for the prevention of malignant tumors are extremely limited and consist of the preventive removal of an organ (while it is still healthy) in which a tumor is expected to appear. After the operation, its function can be replaced. For carriers of highly penetrant BRCA1/2 mutations, bilateral salpingo-oophorectomy or more frequent screening is recommended. This translates to a reduction in OC and BC-specific mortality [38]. PARP inhibitors, furthermore, were added to the treatment regimens of these patients.

## 4. Materials and Methods

### 4.1. Patients and Their Histories

In our study, we collected all essential information about 60 female patients with PMMNs who had taken a part in the research: age of manifestation, interval between first and subsequent malignancies, their localizations, treatments, and family cancer histories.

Each of them received treatment for malignant neoplasias in the Novosibirsk Region Clinical Oncology Center or in E. Meshalkin National Medical Research Center of the Ministry of Health of the Russian Federation, Novosibirsk. Each patient had a history of two or three malignant neoplasms and had developed BC, OC, or both. Only patients who lived all their lives in the Novosibirsk region were included in the study. Therefore, the environmental impact (in particular, the influence of background radiation and insolation) was nearly the same for all patients.

The family cancer history was drawn up for concrete patients; information about their relatives’ cancer histories was assembled. The criterion for classifying a family history as aggravated was the presence of two or more tumors from the HBOC spectrum in first-degree, second-degree, and third-degree relatives (parents, grandparents, siblings, uncles, and aunts).

### 4.2. Morphological Examination

The material for the morphological study was obtained during the cytoreductive surgery. The morphological examination was carried out using light-optical microscopy (Zeiss Axio-Imager.M2 microscope (Carl Zeiss Microscopy GmbH, 07745, Jena, Germany)) with staining of sections by hematoxylin and eosin.

### 4.3. Targeted Sequencing

Then, we estimated the frequency of pathogenic variants in eight locations: *BRCA1* 185delAG (p.Glu23fs; rs80357914), *BRCA1* T300G (p.Cys61Gly; rs28897672), *BRCA1* 2080delA (p.Tyr655fs; rs80357522), *BRCA1* 4153delA (p.Glu1346fs; rs80357711), *BRCA1* 5382insC (p.Gln1756fs; rs80357906), *BRCA2* 6174delT (p.Ser1982fs; rs80359550), *CHEK2* 1100delC (p.Thr367fs; rs555607708), and *BLM* C1642T (p.Gln548Ter, rs200389141) for women who had developed PMMNs.

From each patient, 2 mL of venous blood was obtained. Samples were collected between November 2017 and April 2018. Ethylenediaminetetraacetic acid was used for safety. Blood samples were stored at −20 °C until further use. The isolation of genomic DNA was performed using the classical phenol-chloroform method.

Isolated DNA was quantified with the QuantumDNA 211 Kit (Evrogen, Moskow, Russia). DNA concentration ranged from 1.1 to 67.4 ng/mL. 

A TruSight Cancer Illumina panel (Illumina Inc, San Diego, CA, USA) was used for the targeted genomic sequencing. Raw sequencing read quality was assessed using FastQC v. 0.11.8 (Babraham Bioinformatics, Babraham Institute, Babraham, Cambridgeshire, UK). The reads were trimmed for quality (less than Q20), and adapters were removed using Trimmomatic. Alignment of reads to the reference human genome GRCh37.75/hg19 (Ensembl) (EMBL’s European Bioinformatics Institute, Hinxton, Cambridgeshire, UK) was performed with a Burrows-Wheeler Aligner (BWA), v. 0.7.17 (by Li H. and Durbin R, Wellcome Sanger Institute, Hinxton, Cambridgeshire, UK). To report alignment statistics and determine read duplicates, we applied Sequence Alignment Map (SAM) tools and Picard tools. Base quality score recalibration was carried out with a Genome Analysis Toolkit 4 (GATK4) (v. 4.1.2) (Broad Institute, Cambridge, MA, USA), and dbSNP (build 144) (National Center for Biotechnology Information, Bethesda, MD, USA). Variant calling was performed with GATK (v. 4.1.2) HaplotypeCaller. We excluded false positives using StrandBiasBySample, StrandOddsRatio, and BaseQualityRankSumTest annotations (Broad Institute, Cambridge, MA, USA), as well as mis-sequenced single-nucleotide variants in polyN motifs, such as GGGTG > GGGGG, CCCCG > CCCCC, and others. For functional annotation of variants, ANNOVAR was used.

## 5. Conclusions

One of the most relevant tasks of clinical oncology is the prediction of new tumors in patients predisposed to cancer. 

Obviously, risk-reducing surgery cannot be applied widely. For most patients, only the early detection of cancer, while it is still curable, is acceptable. To predict the localization of a tumor, the clinician needs to know about the presence of oncogenic mutations in the patient’s genome, his family cancer history, and the features of already developed neoplasm(s). The integrated use of clinical and laboratory methods for examining patients, including targeted sequencing, can bring success in the early diagnosis and personalized treatment of tumors in patients with PMMNs and their relatives. This approach can increase the length and quality of their lives.

## Figures and Tables

**Figure 1 ijms-24-06705-f001:**
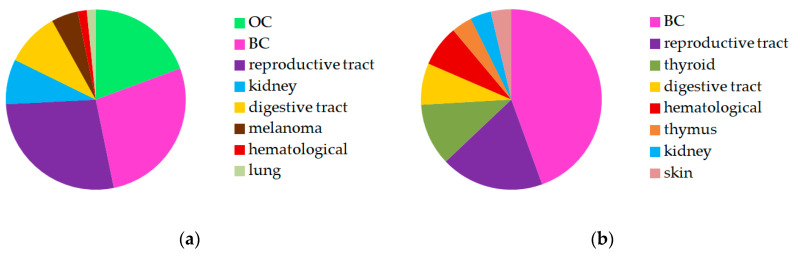
The other invasion sites: (**a**) in patients with BC; (**b**) in patients with OC. Three combinations per case are considered for triple primary malignancies.

**Figure 2 ijms-24-06705-f002:**
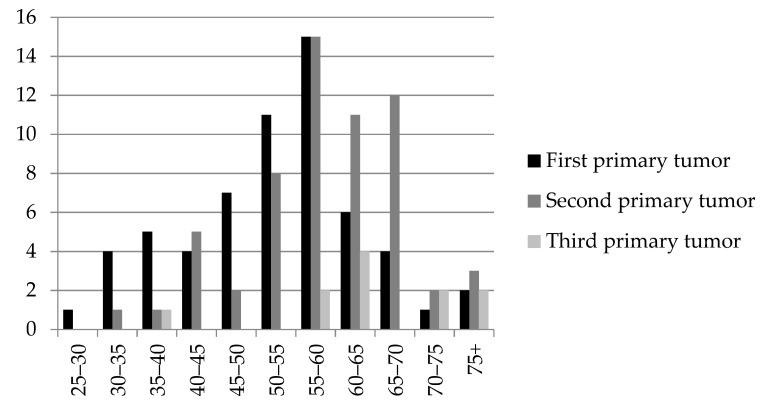
Ages of patients at the time malignant tumors were first detected.

**Figure 3 ijms-24-06705-f003:**
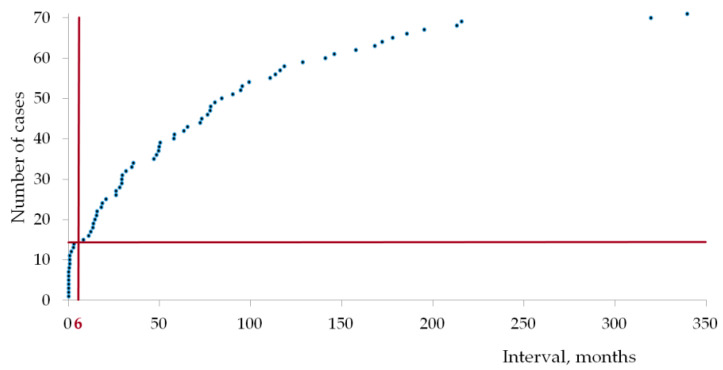
Intervals between the previous and the next malignancy. Two intervals per case are considered for triple primary malignancies.

**Figure 4 ijms-24-06705-f004:**
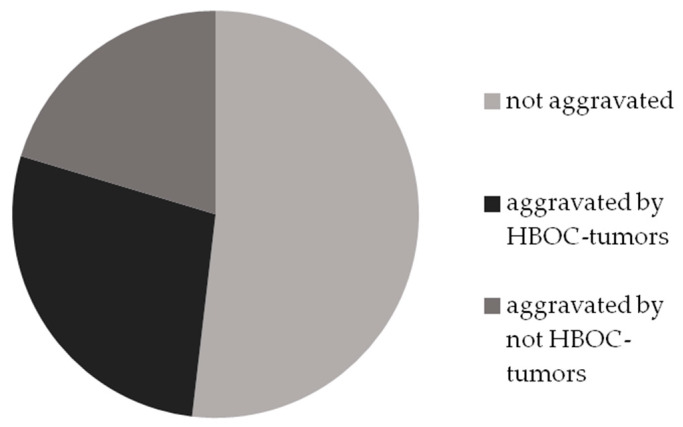
Aggravation of family histories by HBOC-spectrum cancers and other malignancies.

**Figure 5 ijms-24-06705-f005:**
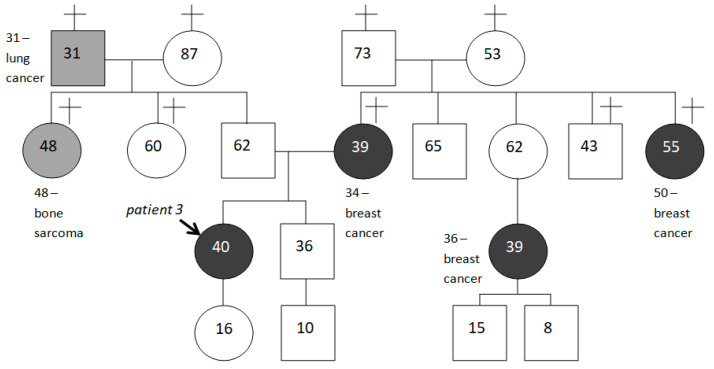
Family history of patient 3, aggravated by tumors of HBOC-spectrum. Persons who developed a tumor of HBOC-spectrum are marked in dark gray; relatives with other malignancies are marked in light gray.

**Figure 6 ijms-24-06705-f006:**
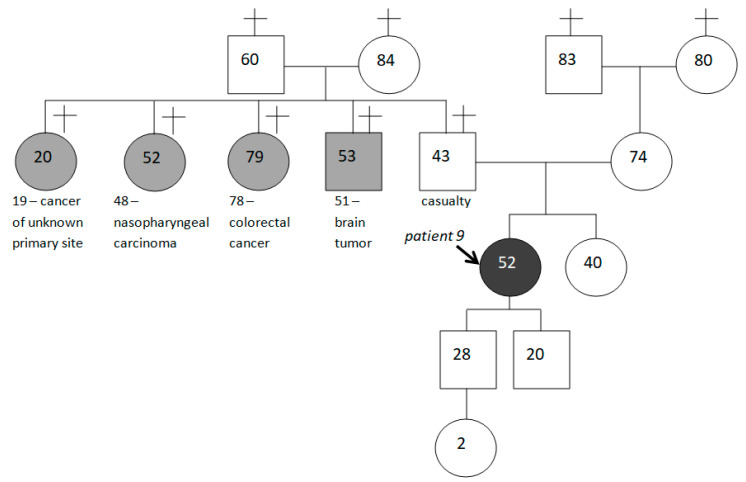
Family history of patient 9, not aggravated by tumors of HBOC-spectrum.

**Table 1 ijms-24-06705-t001:** Mutation carriers and their family histories.

	The First Tumor			The Second Tumor			The Third Tumor			
Case	Histological Type	TNM Stage	Age	Histological Type	TNM Stage	Age	Histological Type	TNM Stage	Age	Family Cancer History: Localization and Age
Patients with *BRCA1* 5382insC										
1.	HGSOC ^1^	T1a N0 M0	55	IBC-NST ^2^ (grade II)	T1a N0 M0	56				PMMNs: BC + OC (up to 50), E(M)C (52)
2.	IBC-NST (grade III)	T1a N0 M0	32	HGSOC	T3 N0 M0	58				CRC ^3^ (70), BC (44), esophageal cancer (37)
3.	IBC-NST (grade II; sin.)	T2 N0 M0	38	IBC-NST (grade III; dex.)	T2 N0 M0	40				BC (34), BC (50), bone sarcoma (48), lung cancer (31)
4.	IBC-NST, (grade II)	T2 N0 M0	48	LG ^4^ serous E(M)C ^5^	T1b N0 M0	60				BC (32), lung cancer (77)
5.	IBC-NST (grade II; sin.)	T2 N0 M0	48	IBC-NST (grade II; dex.)	T2 N0 M0	59	HGSOC	T1b N0 M0	60	stomach cancer (50)
6.	IBC-NST (grade II; sin.)	T1 N0 M0	54	IBC-NST (grade III; dex.)	T1 N1 M0	59	HGSOC	T3c N0 M0	64	BC (48), BC (60)
7.	BC	T2 N1 M0	32	IBC-NST (grade II)	T1a N0 M0	50	Uterine leiomyosarcoma	T2 N0 M0	62	Bilateral BC (41); thyroid cancer (52)
Patient with *BRCA1* T300G										
8.	IBC-NST (grade II; sin.)	T2 N1 M0	59	IBC-NST (grade II; dex.)	T2 N1 M0	63				BC (60), stomach cancer (74), prostate cancer (52)
Patient with *BRCA1* 4153delA										
9.	IBC-NST (grade II)	T2 N1 M0	49	HGSOC	T3 N0 M0	52				CRC (78), cancer of unknown primary site (19), malignant brain tumor (51), nasopharyngeal carcinoma (48)

^1^ HGSOC—high-grade serous ovarian carcinoma; ^2^ IBC-NST—invasive breast carcinoma of no special type; ^3^ CRC—colorectal cancer; ^4^ LG—low-grade; ^5^ E(M)C—endometrial cancer (carcinoma).

## Data Availability

Epidemiological data for Russia and the Novosibirsk region are taken from open annual reports of the Ministry of Health: The state of oncological care for the population of Russia (from 2011 to 2021).

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
