# Peer review of "Genetic Analysis of Multiple Primary Malignant Tumors in Women with Breast and Ovarian Cancer"

_ijms, 2023, doi:10.3390/ijms24076705_

Round 1

Reviewer 1 Report

The manuscript entitled “Genetic analysis of multiple primary malignant tumors in women with breast and ovarian cancer” is nice study and good focus area. Authors have screened eight germline mutations using targeted genomic sequencing which provoke the majority of cases of hereditary breast and ovary cancer syndrome (HBOC). However, the manuscript needs to be improve significantly before publication. A major revision is needed to improve the quality of this work which is as follows: 

1. Authors should use either abbreviation after mentioning once. For example, the sentence “in the primary malignant neoplasias (PMMNs) is firstly mentioned with abbv. Which is again used as full form in the manuscript. Please check and make it uniform. Please indicate (full form) what is PARP-inhibitors.

2. The brief development procedure of PMMNs should precisely discuss in the first part of the introduction.

3. The introduction part needs to be improved significantly and some sentences seems like incomplete. There should be a last paragraph in the introduction discussing what has been done in the manuscript. 

4. More references need to cite in the introduction part mentioning some advanced screening technologies. Some important references need to be addedare Mater. Today. Chem., 26, (2022), 101182  and Biomater. Sci., 2022, 10, 5054-5080

5. The materials and methods section should be well categorized. The patient’s characteristics part needs to be merged with the provided information of the patients. 

6. Morphological examination was carried out using light-optical microscopy with staining of sections by hematoxylin and eosin. Model number of the equipment should be mentioned. 

7. In the dicussion part, all the possible factors for PMMNs should be arranged properly such as age, sex hormones, environmental factors, family history with one one paragraph to improve the reader’s knowledge. 

8. The authors should clearly mention the gene mutation types occuring with particular external factors in the dicussion part. Authors have mentioned “Only 5 (â„–1, 3, 6, 7, 8) out of 9 of family histories were aggravated by tumors of HBOC-spectrum.” But the information about patient 3, and 6 are missing. As  mentioned “Patients 4 and 7 also had not-HBOC tumors” needs to be clarified, referring to the previous statement made.

9. The arrangement of the paragraphs is very poor. English grammar polishing is needed.

10. A summary or conclusion mentioning what is the outcome of this study needs to add to the manuscript, which will improve the reader’s prospects towards this study. 

Author Response

Thank you very much for such a comprehensive review. To improve certain sections of the manuscript, new paragraphs and references have been added.

  1. Authors should use either abbreviation after mentioning once. For example, the sentence “in the primary malignant neoplasias (PMMNs) is firstly mentioned with abbv. Which is again used as full form in the manuscript. Please check and make it uniform. Please indicate (full form) what is PARP-inhibitors.

We have checked all abbreviations and made them uniform, indicated the full form for PARP-inhibitors (poly(ADP-ribose)polymerase inhibitors) and explained when they are recomended to be used.

2. The brief development procedure of PMMNs should precisely discuss in the first part of the introduction.

Description of the development of PMMNs has been added to the introduction.

3. The introduction part needs to be improved significantly and some sentences seems like incomplete. There should be a last paragraph in the introduction discussing what has been done in the manuscript. 

We have made the introduction more complete and added the paragraph discussing what has been done to it.

4. More references need to cite in the introduction part mentioning some advanced screening technologies. Some important references need to be addedare Mater. Today. Chem., 26, (2022), 101182  and Biomater. Sci., 2022, 10, 5054-5080

Thank you for the links to these important articles describing the development and application of modern technologies in medicine. The information they contain is extremely interesting. We hope that these methods will soon be widely used in practice. We have added the first of these articles to the references. But we consider the second one (about the three-dimensional bioprinting) to be far from the theme of our study and don't find it possible to add it. 

5. The materials and methods section should be well categorized. The patient’s characteristics part needs to be merged with the provided information of the patients. 

We have structured the Materials and Methods part by dividing it into three subsections:

Patients and their histories;

Morphological examination;

Targeted sequencing.

We have indicated in this section only the criteria for inclusion of patients in the study. For example, we deliberately included only patients who have lived in the Novosibirsk region all their lives, and not others. This was done to eliminate environmental contributions that patients may have been exposed to in other geographic areas.

6. Morphological examination was carried out using light-optical microscopy with staining of sections by hematoxylin and eosin. Model number of the equipment should be mentioned.

We have indicated the model number of the equipment used. 

7. In the dicussion part, all the possible factors for PMMNs should be arranged properly such as age, sex hormones, environmental factors, family history with one one paragraph to improve the reader’s knowledge.

We have added several paragraphs to the discussion part, which list all the important risk factors leading to the initiation and progression of PMMNs.

8. The authors should clearly mention the gene mutation types occuring with particular external factors in the dicussion part. Authors have mentioned “Only 5 (â„–1, 3, 6, 7, 8) out of 9 of family histories were aggravated by tumors of HBOC-spectrum.” But the information about patient 3, and 6 are missing. As  mentioned “Patients 4 and 7 also had not-HBOC tumors” needs to be clarified, referring to the previous statement made.

We have added the paragraph describing the causes of the occurrence of mutations/tumor initiation. We also noted that new mutations occur at random locations in genomic DNA.  These locations aren't related to causes (which included replication mistakes, and DNA damage by radiation and carcinogenic substances). 

All the nessesary information about mutation carriers: ages of tumors' onest, localizations, histological features of malignancies, and family cancer histories, is presented in the Table1. 

Please let us know if there is anything else we should clarify.

9. The arrangement of the paragraphs is very poor. English grammar polishing is needed.

We have changed the arrangement of the paragraphs where possible.

10. A summary or conclusion mentioning what is the outcome of this study needs to add to the manuscript, which will improve the reader’s prospects towards this study.

The conclusion part has been added to the manuscript.

Reviewer 2 Report

The paper of  Alina Savkova et al. to be published the minor revisions:

-For greater clarity, it is requested to divide the section of materials and methods into different subsections for each technique used

-It was suggested that a paragraph be separated from the conclusions from the discussions in order to clarify them

Author Response

Thank you very much for a review. Some changes have been made to the manuscript to improve it.

  • We have structured the Materials and Methods part by dividing it into three subsections:

    Patients and their histories;

    Morphological examination;

    Targeted sequencing.

  • The conclusion part has been added to the manuscript.

Round 2

Reviewer 1 Report

Accept as it is. The authors have improved the manuscript well.